# Does Structural Color Exist in True Fungi?

**DOI:** 10.3390/jof7020141

**Published:** 2021-02-16

**Authors:** Juliet Brodie, Colin J. Ingham, Silvia Vignolini

**Affiliations:** 1Department of Life Sciences, Natural History Museum, London SW7 5BD, UK; j.brodie@nhm.ac.uk; 2Hoekmine BV, 3515 GJ Utrecht, The Netherlands; 3Department of Chemistry, University of Cambridge, Cambridge CB2 1EW, UK; sv319@cam.ac.uk

**Keywords:** Myxomycetes, iridescence, pigmentation, evolution of color, mycelia, cell organization, living photonics

## Abstract

Structural color occurs by the interaction of light with regular structures and so generates colors by completely different optical mechanisms to dyes and pigments. Structural color is found throughout the tree of life but has not, to date, been reported in the fungi. Here we give an overview of structural color across the tree of life and provide a brief guide aimed at stimulating the search for this phenomenon in fungi.

The rich diversity of the color palette throughout the tree of life is usually obtained via dyes and pigments, with the fungal kingdom (*Eumycetes*) alone producing a vast array of hues [1]. However, most vibrant colors and color effects found in nature are created using transparent materials with architecture on the same scale as the wavelength of visible light (a few hundred nanometers). Such coloration, so-called structural color, is based on light interference phenomena and is usually independent from the chemical composition of the material. This is in contrast to pigments, whose color depends on light absorption dictated by their molecular composition. [2]. Structural color has been known since Hooke and Newton, who in the 17th century, studied how the peacock and house fly generated intensely colored surfaces without extractable pigments [3]. Four centuries later, the optics of structural color in life is now a growing field, and it is clear that multiple types of structural color are found widely in the eukaryotes [4,5,6]. Examples of structural color are found in red, green, and brown macroalgae [7,8], but also in different plant tissues: from leaves to flowers to fruits (Figure 1) [9,10]. The functions of structural color in both plants and algae are still not fully understood, and depending on the system considered, can span from light management, including improving photosynthesis and photoprotection [11], to interspecies communication [9]. Similarly, structural color is also extremely widespread within the invertebrates, (such as insects, arachnids, and marine cephalopods [4,5,6]), and vertebrates, such as birds, and reptiles including mammals [2,4,12,13,14,15]. Demonstrable advantages of structural color for animals are largely connected with visual properties and include intra- and interspecies signaling and camouflage [2,4]. 

In all cases, for a living organism to create structural color a formidable capacity is required for biological organization on the nanoscale, posing questions as to the evolution of structural organization and color.

Structural color also exists in microorganisms and is well-known within the Myxomycetes (Figure 2), a phylogenetic group distinct from true fungi but with many similarities, including sporulation and saprotrophy [16,17]. Structural color in the Myxomycetes (often multihued and pointillistic) is due to thin-film interference caused by light interacting with the peridium, the thin transparent membrane which encloses clusters of asexual spores [16]. Within the prokaryotes, we also find examples of structural color, albeit not as individual cells, but in bacteria colonies. Gliding bacteria such as *Flavobacterium* IR1 (Figure 3) and *Cellulophaga lytica* are known to produce vivid and iridescent coloration due to the periodic organization of their cells [18,19,20,21].

Therefore, considering that such structural colors are so widespread in a great variety of living organisms, it is puzzling that structural colors have never been reported in the Eumycetes. The fungi are a structurally and chromatically sophisticated kingdom, and yet structural color appears, to the best of our knowledge, to have never been reported. From an evolutionary point of view, it would be fascinating if structural color remains to be discovered within the fungi but also notable if genuinely absent. 

Structural colors can be subtle to observe and often the optical effects are poorly described in papers and therefore are probably under-reported. Unfortunately, there is no substantive genomics information that would enable a bioinformatics approach. There are almost no genes relating to structural color known in eukaryotes except for the *optrix* gene which has been identified as relevant to both pigmentation and structural effects in butterflies [22]. Even in a well-known organism such as the peacock, with a fully sequenced genome, the genes encoding the nanoscale architects responsible for the vivid colorations displayed by the feathers remain obscure [23]. Further, whilst some genes specifying structural colors are known in bacteria [18], it is not clear if there is any direct relevance to the fungi. 

Discriminating between color produced by pigments and structural color can require a combination of spectroscopical and anatomical studies [24], often accompanied by modeling efforts. This is due to the fact that often structural colors are combined with coloration effects provided by pigments, making it harder to disentangle and differentiate these two distinct optical mechanisms. In practice, given most labs do not have these resources, there are a number of simple tests that could be useful to the mycologist in determining if an isolate might have structural colors. Firstly, many forms of structural color are best observed using a strong, directed white light, i.e., direct sunlight, a broad-spectrum LED, or an incandescent bulb. Viewing under diffuse illumination, such as fluorescent strip lighting, makes it more challenging the observation of structural colors as this can enhance the coloration produced by the pigments. Secondly, the following checklist may be useful in trying to decide if a fungus is structurally colored:(1)Does the color shift with a change in viewing and/or illumination angle (Figure 3a,b);(2)If you disturb the structure by mixing or grinding, does the color vanish (Figure 3a);(3)Are there punctuated, extremely saturated colors under direct white light (Figure 3a,c);(4)Does the color reversibly change (or appear/disappear) if the material is immersed in water or other solvents;(5)When viewed by electron microscopy, is there a degree of ordering into repetitive units with each subunit on the hundreds of the scale of a few hundred nanometers?

If the answer to one or more of these questions is “yes” then structural color may be involved.

We hope that mycologists will be stimulated to search for colors beyond pigmentation and would be happy to assist in this quest.

## Figures and Tables

**Figure 1 jof-07-00141-f001:**
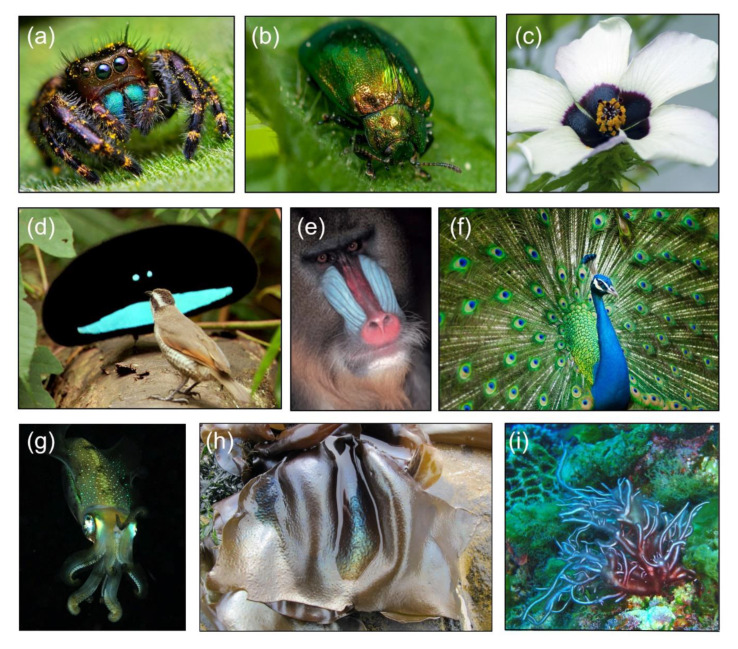
Examples of structural color in eukaryotes. (**a**) Female jumping spider *Phidippus audax* (Photo: T. Shahan). (**b**) Dogbane beetle, *Chrysochus auratus* (Photo: P. Ganai). (**c**) *Hibiscus trionum* flower (Photo: E. Moyroud). (**d**) Male superb bird of paradise, *Paradisaea rudolphi*. (**e**) Male mandrill, *Mandrillus sphinx*. (**f**) Male Indian peafowl (peacock), *Pavo cristatus*. (**g**) Bigfin reef squid (or glitter squid), *Sepioteuthis lessoniana*. (**h**) Red macroalgae, *Iridaea cordata* (Photo: J. Brodie). (**i**) Red macroalga, *Trichogloea* spp. (Photo: G. Saunders).

**Figure 2 jof-07-00141-f002:**
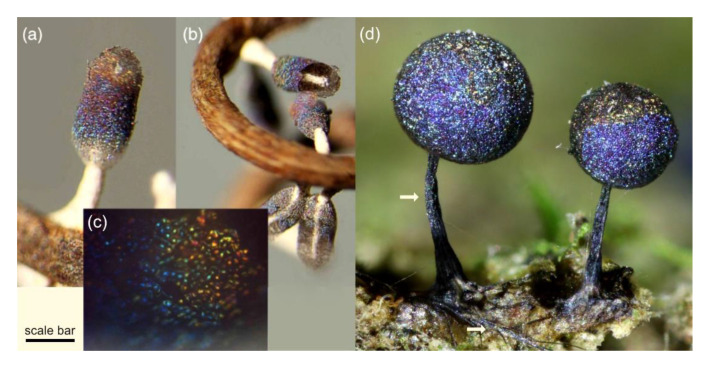
Examples of structural color in the Myxomycetes. (**a**–**c**) *Diachea leucopoda* (Photo: M. Inchaussandague) with spores enclosed within the transparent peridium, displaying bright, iridescent, structural color. (**c**) A close up of the surface of the sporulating body, with individual spores visible. (**d**) *Lamproderma* sp. (Photo: S. Lloyd) with iridescent sporulating bodies enclosed within the peridium but also iridescence visible on the stalk and hypothallus (e.g., white arrows). Scale bars: (**a**) = 1.2, (**b**) = 2, (**c**) = 0.3, (**d**) = 0.2 mm.

**Figure 3 jof-07-00141-f003:**
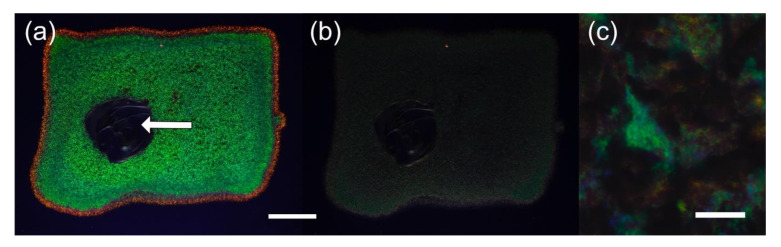
Structural color of *Flavobacterium* IR1 cultured on nutrient agar containing a black dye for contrast. (**a**) Structural color viewed at the optimal angle. Arrow indicates where the cells were mixed with an inoculation loop, disrupting the photonic crystal and so the coloration. (**b**) As panel (**a**) but viewed from a suboptimal angle for structural color. (**c**) Close up of structural color from panel (**a**) showing that what appears green to the eye is actually a pointillistic arrangement of multiple colors. Scale bars: (**a**) and (**b**) = 6 mm, (**c**) = 20 µm.

## Data Availability

Not applicable.

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
