# Peer review of "Does Structural Color Exist in True Fungi?"

_jof, 2021, doi:10.3390/jof7020141_

Round 1

Reviewer 1 Report

I found this a very engaging and interesting note, especially as a mycologist who works specifically with fungal pigments. I have no suggested edits, and my only comment is that I am now very intrigued! I think this is well worth publishing.

I wonder how the authors would classify 'blue stain' fungi? These fungi secrete a dark brown/black melanin pigment, but in wood, this pigment interacts with light in some cases, at some angles, to appear blue--varying shades, from sky blue to a muted grey. This of course depends on the wood structure as well, and seems likely to be just a pigment interacting with wood, but based on the information presented in the letter now I wonder if it doesn't have some other properties. 

Author Response

Thank you for the supportive review. On blue stain fungus,  this certainly is interesting though we could not tell more from the images we viewed. We found this an encouraging comment, as it suggests the article has already elicited a desired response, and will look at 'blue stain' fungi as possible candidates for structural colour.

Reviewer 2 Report

The article talks about the existence of Structural coloring in fungi. The topic is interesting since structural coloring is a very complex feature typical of Eukaryotes, but found also bacteria. Unfortunately, there are not articles showing structural color in Eumycetes, but is interesting to talk about it.

Author Response

Thank you for the review, we reread the article and made minor grammatical improvements and checked the spelling, as instructed.